# Exploring Transcriptomic Landscapes in Red Blood Cells, in Their Extracellular Vesicles and on a Single-Cell Level

**DOI:** 10.3390/ijms232112897

**Published:** 2022-10-25

**Authors:** Erja Kerkelä, Jenni Lahtela, Antti Larjo, Ulla Impola, Laura Mäenpää, Pirkko Mattila

**Affiliations:** 1Finnish Red Cross Blood Service, Kivihaantie 7, 00310 Helsinki, Finland; 2Institute for Molecular Medicine Finland (FIMM), University of Helsinki, Tukholmankatu 8, 00290 Helsinki, Finland; 3Research Programs Unit, Molecular Neurology, University of Helsinki, Tukholmankatu 8, 00290 Helsinki, Finland

**Keywords:** erythrocyte, MALAT1, RNA sequencing, transcriptomics, extracellular vesicle

## Abstract

Being enucleated, RBCs lack typical transcriptomes, but are known to contain small amounts of diverse long transcripts and microRNAs. However, the exact role and importance of these RNAs are lacking. Shedding of extracellular vesicles (EVs) from the plasma membrane constitutes an integral mechanism of RBC homeostasis, by which RBCs remove unnecessary cytoplasmic content and cell membrane. To study this further, we explored the transcriptomes of RBCs and extracellular vesicles (EVs) of RBCs using next-generation sequencing. Furthermore, we performed single-cell RNA sequencing on RBCs, which revealed that approximately 10% of the cells contained detectable levels of mRNA and cells formed three subpopulations based on their transcriptomes. A decrease in the mRNA quantity was observed across the populations. Qualitative changes included the differences in the globin transcripts and changes in the expression of ribosomal genes. A specific splice form of a long non-coding RNA, Metastasis Associated Lung Adenocarcinoma Transcript 1 (MALAT1), was the most enriched marker in one subpopulation of RBCs, co-expressing with ribosomal structural transcripts. MALAT1 expression was confirmed by qPCR in CD71-enriched reticulocytes, which were also characterized with imaging flow cytometry at the single cell level. Analysis of the RBC transcriptome shows enrichment of pathways and functional categories required for the maturation of reticulocytes and erythrocyte functions. The RBC transcriptome was detected in their EVs, making these transcripts available for intercellular communication in blood.

## 1. Introduction

Human erythrocytes are specialized in providing gas transport throughout the body. Complex genetic networks control hematopoietic stem cell differentiation into progenitors that give rise to billions of erythrocytes daily. Erythroblasts undergo consecutive maturation steps, culminating in enucleation to yield reticulocytes and subsequently, biconcave erythrocytes [1]. The overall process through which the reticulocyte is converted to an erythrocyte is not completely understood. Early reticulocyte maturation is characterized by extensive membrane remodeling and the selective removal of unnecessary plasma membrane proteins (e.g., CD71, CD98) through vesiculation via the endosome-exosome pathway [2,3]. In contrast, late maturation is characterized by the degradation and elimination of the final cell organelles through autophagy and mitophagy. These processes enclose unwanted materials into autophagosomes and remove them from the cell [4,5,6].

During terminal erythroid differentiation, globin mRNAs accumulate and comprise over 95% of total cellular mRNA [7]. Retained ribosomes and pre-synthesized mRNAs permit continued translation activities in reticulocytes after enucleation, especially that of hemoglobin (Hb) [8,9,10,11]. Apart from Hb mRNA, reticulocytes have been shown to contain surprisingly diverse transcriptomes. These transcriptomes are hypothesized to encode information regarding terminal erythroid differentiation and maturation [9,11,12]. However, the role and regulation of these transcripts in these final stages of reticulocyte maturation is largely unknown, even though non-coding RNAs are known to have an impact on erythroid differentiation in general [13].

The remarkable enrichment of globin mRNAs relies on their exceptionally long half-lives and possibly also selective degradation of non-globin mRNAs [8,14]. Enucleated red blood cells (RBCs) are unable to transcribe new mRNA. Thus, a progressive decrease in all cellular RNA is observed during their maturation. The disappearance of RNA on a large scale occurs simultaneously with the loss of cellular structures such as the mitochondria and ribosomes [15]. Multiple mechanisms of mRNA stability and degradation have been described for α-globin, β-globin as well as other mRNAs [16,17,18,19,20].

Extracellular vesicles (EVs) are a heterogeneous population of phospholipid membrane particles that may contain functional molecules, such as RNA and have a well-established role in intercellular signaling [21,22]. RBCs, together with platelets, are the main EV-secreting cells in the blood [23]. In addition to having an important role in reticulocyte maturation, vesiculation is proposed as one mechanism that allows erythrocytes to clear away damaged or harmful molecules and thus prevent their early removal from circulation. Similar vesiculation occurs during storage-related aging [24,25]. The RNA content of EVs is of special interest, especially from the point of view of biomarker discovery [26]. To our knowledge, no one has analyzed the transcriptomes of EVs originating from RBCs.

We studied the transcriptomes of single-cell RBCs from both fresh blood and RBC unit. In parallel, we used RNA sequencing to study bulk transcriptomes of RBCs and EVs isolated from RBC units. Our study shows that although the overall RNA content in RBCs is low, subpopulations of RBCs have significant differences in their transcriptomes, which are especially defined by globin transcripts and expression of a long non-coding RNA (lncRNA), namely, Metastasis Associated Lung Adenocarcinoma Transcript 1 (MALAT1).

## 2. Results

Four leukocyte-free RBC units were used to analyze transcriptomes in RBCs and RBC-derived EVs (EV characterization is described in Appendix A). In parallel, RBC transcriptomes from one RBC unit and one fresh blood sample were analyzed on a single-cell level to explore subpopulations of RBCs (Figure 1).

### 2.1. RBC Transcriptome Is Scarce but Relatively Rich and Is Transferred to RBC-Derived EVs

For transcriptomic analysis, the mean amounts of uniquely aligned reads produced for the four RBC samples and four EV samples were 39.6 × 10^6^ and 21.7 × 10^6^, respectively (Appendix A). The RNAseq data quality was good as judged by body read coverage and principal component analyses (Appendix A). In general, according to the correlation plots, the diversity of transcriptomes among the EV group was higher than among the RBC group (Appendix A).

An abundant repertoire of expressed genes was found in all RBC samples (Appendix A). ‘RBC transcriptome’ used for subsequent annotations, as well as functional category enrichment and pathway analyses, consists of 2070 (40%) expressed genes that were shared among all 4 samples (Figure 2A). Corresponding EV samples contained fewer expressed genes (Appendix A) and were more diverse in the RNA content, as only 686 (15%) of the expressed genes were shared in all samples (Figure 2B). Most of the transcripts (555 out of 686) in EVs, were also found in RBCs. In addition, most highly expressed genes were shared between RBC and EVs, such as well-known erythrocyte biology-related gene hemoglobin subunit beta (HBB, vast majority of transcripts), ferritin (FTL), hemogen (HEMGN), and 5’-aminolevulinate synthase 2 (ALAS2). The remaining 131 transcripts in EVs (highlighted in yellow in Appendix A) that were not found in RBCs comprise probable contamination of RNA from EVs of other cells, mostly platelets, possibly also leukocytes and endothelial cells due to a small residue of plasma in the blood unit production process (maximum 20 mL/RBC unit). However, in western blot analysis, we did not detect platelet marker CD61 (Appendix A). Instead of contamination markers, RBC marker CD235a and Hb were clearly visible in Western blot, indicating that most of the EVs are of RBC origin. The lack of leukocyte or platelet transcripts in the RBC data confirmed that the RBC transcriptome is of RBC origin.

### 2.2. Functional Category and Canonical Pathway Analyses Reveal Processes Related to RBC Maturation and Function

Among the most enriched canonical pathways in RBCs were eukaryotic initiation factor 2 (eIF2) signaling and its regulation, mitochondrial dysfunction, protein ubiquitination pathway, oxidative phosphorylation, and mechanistic target of rapamycin (mTOR) signaling (Figure 3A and Table 1, Appendix A). The same pathways were also enriched in EVs, of which eIF2 signaling and mitochondria-related pathways had even lower *p*-values than RBCs (Figure 3B, Table 1, Appendix A). Different enrichment in pathways was also evident between RBCs and EVs, such as heme biosynthesis II, NRF2-mediated oxidative stress response- and hypoxia signaling -pathways in RBCs with very significant *p*-values, while two latter were not statistically significantly enriched in EVs.

The top shared molecular and cellular functions in RBC and EV transcriptomes were related to gene expression, protein synthesis, RNA modification, and cell death and survival (Table 1, Appendix A). Under the ‘physiological system development and function’ –category, ‘hematological system development’ and ‘hematopoiesis’ were among the top five most enriched categories as expected, with several erythroid cell-related subcategories being the most significant (Table 1, Appendix A). Other obvious RBC functions included synthesis and homeostasis of different RBC–related molecules, such as porphyrin, heme, and iron (category ‘small molecule biochemistry’, *p*-values 7.61 × 10^−8^–7.3 × 10^−5^ in RBCs) and free radical scavenging, which was a more significant category in EVs compared to RBCs (Table 1, Appendix A). Furthermore, the most significant subcategory under ‘cellular function and maintenance’ was autophagy (*p*-value 1.26 × 10^−11^ in RBCs), strongly related to reticulocyte maturation.

Suspected contamination present in the EV transcriptome is reflected by the presence of ‘inflammatory response’ and ‘immune cell trafficking’ -categories and different leukocyte–related subcategories in, e.g., ‘hematological system development’ (Appendix A). Due to contamination, the differential expression analysis between RBC and EV transcriptomes was not performed. However, it was still evident that more mitochondrial transcripts were present in the EV transcriptome, as judged by expression intensity and diversity (Appendix A). That is most likely why mitochondria-related canonical pathways had considerably more significant *p*-values in EV transcriptome compared to that of RBCs (Table 1).

### 2.3. Sequencing on the Single-Cell Level Reveals Separate RBC Subpopulations Defined by Differential Expression of Hemoglobin and the Long Non-Coding RNA, MALAT1

To study the cellular heterogeneity among RBCs, we analyzed fresh and blood unit RBCs using the single-cell transcriptomics method from 10X Genomics [27], aiming to capture 2000 cells per sample. A total of 396 RBCs in blood unit and 173 RBCs in fresh blood samples were initially judged as cells based on the total number of detected transcripts. Total number of detected genes per sample was 1142 for blood units and 992 for fresh RBCs. As expected, the number of captured transcripts, detected as unique molecular identifiers (UMIs), varied greatly between individual cells (4–4201 for blood unit and 12–2258 for fresh RBCs, mRNA capture efficiency of our single-cell instrument being 10–15%), and the median number of genes detected per cell was low (5 for blood unit and 15 for fresh RBCs).

A total of 554 RBCs from the blood unit and fresh blood were used for the final analysis. The genes with 5 or more total transcripts across the cells (158 in total, Appendix A) included 122 (77%) shared genes with bulk RBC transcriptome (Appendix A). Among these were genes known to be associated with RBC biology such as hemoglobin subunit beta (HBB), hemoglobin subunit alpha 1 and 2 (HBA1/HBA2), ferritin light chain (FTL), and glycophorin C (GYPC). This finding demonstrated the consistency between the RNA sequencing analysis methods used.

The RBCs were further analyzed to identify subpopulations. The cells from the blood unit and fresh blood samples clustered in an overlapping manner, supporting our decision to use combined analysis with merged data (Figure 4A). Clustering of the RBCs based on their transcriptome profiles resulted in three distinct cell clusters (Figure 4B). Cells in cluster 1 had the highest amount of detected total transcripts (AvgUMI = 110), while in clusters 0 and 2 the total transcript counts were lower (AvgUMI = 60 and 65, respectively) (Figure 4C,D). This could mean that cells in different clusters may represent RBCs, more specifically reticulocytes, at various maturation stages. The variation measured by the number of genes detected was highest in clusters 1 and 0, while the percentage of mitochondrial genes was highest in cluster 0 (Figure 4D).

Cells in cluster 1 were characterized by a high expression of RBC-specific genes such as HBB, HBA1, HBA2, FTL, and GYPC (p < 0.0001 for all) (Figure 5A–C, Appendix A), suggesting that these cells most likely represent the population of young reticulocytes, while the cells in clusters 0 and 2 could represent more mature reticulocytes or early erythrocytes.

The cells in cluster 0 were significantly enriched with long non-coding RNA MALAT1 transcripts (p < 0.0001) (Figure 5A–C, Appendix A). Cluster 0 was also characterized by the low or absent expression of HBB, HBA1, and HBA2, and it was enriched in the expression of mitochondrial and ribosomal genes such as mitochondrially encoded cytochrome c oxidase I and II (MT-CO1 and MT-CO2), ribosomal protein S19 (RPS19) and ribosomal protein S18 (RPS18) (p < 0.0001 for all, Figure 5A, Appendix A). Additionally, few genes typically expressed in platelets, including beta-2-microglobulin (B2M), thymosin beta 4, X-linked (TMSB4X), and thymosin beta 10 (TMSB10), were detected in cluster 0 cells.

Cells in cluster 2 were characterized by very low total UMI count and expression of HBB, HBA2, and FTL genes (p < 0.0001 for all, Figure 5A, Appendix A). The single-cell analysis also provided further evidence for the purity of the RBC population. In case of contamination with leukocytes, they would have formed a separate cluster characterized by relevant markers.

### 2.4. All Reads for MALAT1 Obtained from Single-Cell Sequencing Map to Short Transcript Forms

According to current annotations in Ensembl Genome Browser, 16 shorter transcripts of MALAT1 are predicted in addition to the well-annotated full-length (8708 bp) transcript MALAT1-202: http://www.ensembl.org/Homo_sapiens/Gene/Splice?db = core;g = ENSG00000251562;r = 11:65497762–65506516, accessed on 9 November 2017.

We used IGV with Ensembl gene annotations to visualize short reads mapping to the MALAT1 gene. Based on the alignment, the sequencing reads did not map to the 3′ end of the full-length transcript MALAT1-202. Instead, they mapped to the three short transcripts MALAT1-203, MALAT1-211, and MALAT1-217 (Figure 6). This finding agreed with the bulk RNAseq data. According to our knowledge, this is the first report of the predicted splice site shared by the transcripts MALAT1-203, MALAT1-211, and MALAT1-217 as an exon 1 start site.

### 2.5. Characterization of CD71-Enriched Reticulocytes

With imaging flow cytometry, we wanted to study the dynamics of RNA content in enriched reticulocytes. Double positive events for CD71 and RNA-binding dye thiazole orange (TO) were gated as CD71^high^, CD71^medium^, and CD71^low-neg^ populations based on intensity values on each channel and images. Of those cells, 35% expressed CD71 at a high level, 33% at a medium level, and 17% had low or negative expression (Figure 7A). Although high CD71 expression generally correlated with high TO intensity, especially in the CD71^high^ population, there were also cells with high CD71 expression and low TO intensity, and vice versa (Figure 7B,C) indicating high diversity in the amount of RNA in reticulocyte subpopulations. CD71-enriched cells had varying morphologies and concave-shaped cells having high CD71 expression and TO intensity were also observed (Figure 7C). Moreover, RBC marker CD235a expression was detected with varying levels of intensity (Figure 7C).

The bulk expression of MALAT1 lncRNA in enriched reticulocytes was confirmed with quantitative real-time PCR (qPCR). As expected, the expression of HBB was high in CD71-enriched cells with Cq values ranging from 15.7 to 20.2 (mean 17.5, SD 1.9), while MALAT1 had a lower but detectable expression with Cq values (mean 32.9, SD 2.2) (Figure 7D).

## 3. Discussion

To understand the final stages of RBC maturation, we studied the transcriptomic landscapes of RBCs and their EVs using traditional RNAseq and single-cell RNAseq. Based on our single-cell analysis, there are significant differences in the transcriptomes of three RBC subpopulations. Two subpopulations are defined by the expression of globin genes, and one subpopulation is defined by the expression of the lncRNA, MALAT1, also expressed in bulk RNAseq data and verified by qPCR.

Indeed, lncRNAs in general, and MALAT1 are dynamically expressed and regulated during erythropoiesis [13,28]. Their depletion severely impaired erythrocyte maturation by inhibiting enucleation [29]. Only some cells in the MALAT1 population contained globin transcripts. Instead, this population was enriched in transcripts coding for ribosomal and mitochondrial structural proteins. We assume that these subpopulations reflect the different stages of reticulocyte maturation and that other transcripts in the MALAT1 -population can only be recovered in the late stage of reticulocyte differentiation when not masked by the bulk globin transcripts and not being bound to organelles due to ceased biogenesis of ribosomes and mitochondria.

During their terminal differentiation in peripheral blood, enucleated reticulocytes adjust their protein expression profiles post-transcriptionally. Although not yet studied in RBCs, there is rapidly accumulating evidence for translational regulation executed by ribosome-associated lncRNAs [30,31]. Our data support this, as we found co-expression of MALAT1 transcripts and a plethora of mRNAs coding for ribosomal structural proteins (Figure 5A) in the same subpopulation of RBCs. Recent footprinting studies have shown that the ribosome is the primary destination for the majority of cytoplasmic lncRNAs [32]. In addition, MALAT1 has also been shown to act as a miRNA sponge and as a potent autophagy inducer [33]. Reticulocytes are known to employ miRNA-based mechanisms that could have a role in both protein translation control and degradation of mRNA during maturation [34,35,36]. Intriguingly, MALAT1 hits from our single-cell data all mapped to the same region, showing predictions of short, cytosolic previously unknown transcript forms of MALAT1, instead of nuclear retained, known long form of MALAT1. Since it is poorly polyadenylated [37], the full-length MALAT1 transcript is unlikely captured in our poly-A-based single-cell assay in RBCs.

An obvious decrease in the mRNA quantity was observed across the subpopulations in single-cell RNAseq data. To support these findings, we employed imaging cell cytometry to correlate cellular RNA content to the expression of reticulocyte marker CD71. A decreasing trend in CD71 expression and intensity of TO staining was observed in CD71-enriched reticulocytes, which is known to indicate reticulocyte maturation [38,39]. However, TO intensity and thus cellular RNA amount, was surprisingly heterogeneous, especially within CD71^medium^ and CD71^low^ populations. Furthermore, we observed varying morphologies across the populations, i.e., also concaved cells having high CD71 and TO intensity. This is partially in contrast to the study by Malleret et al. who concluded that CD71^high^ and CD71^medium^ reticulocytes share a similar gross morphology (large and multilobular) when compared to the smaller and increasingly concave reticulocytes seen in the CD71^low^ and CD71^neg^ populations [39], however, their imaging method was different from ours.

In our bulk RBC transcriptomes, both the number of expressed genes and the selection of highly expressed genes match well with the previously published ones [9,11,12] despite the different experimental setups used. The RBC and EV transcriptomes reflect the processes needed both in the reticulocyte maturation as well as for mature erythrocyte functions. During the two days in circulation, reticulocytes need to balance the production of Hb and the final termination of protein translation processes. The enrichment of both the protein synthesis category and the mTOR pathway in our data reflect the uniquely high demand for Hb protein translation, supported by the finding that inhibition of mTOR resulted in reduced maturation of reticulocytes [40]. Additionally, the enrichment of eIF2 signaling indicates active protein translation. In contrast, the enriched ubiquitin-proteasome pathway and ubiquitination category also found to be enriched by Doss et al. [12], points to the targeted degradation of unnecessary proteins in reticulocytes [41] and ubiquitination is an essential step both in enucleation [42] and in mitochondrial autophagy [43]. Interestingly, a recent proteomics study by Chu et al. comparing CD71-positive and -negative RBCs revealed the same set of canonical pathways upregulated during reticulocyte maturation as our transcriptomic level data [44].

Autophagy was a highly enriched process in the RBC transcriptome and is crucial for the removal of mitochondria and ribosomes at the final maturation stage [45]. Mitophagy can occur in reticulocytes through the autophagy-related gene (ATG) protein pathway [45] and through an ATG-independent mechanism involving BCL2 interacting protein 3 like (BNIP3L) [46,47], which was one of the highly expressed transcripts in RBCs (bulk and single-cell data) and EVs, supporting the finding of Doss et al. [12]. Players from both of these pathways were present in our data, except for the lysosomal compartment. According to the theory of Griffits and co-workers, autophagosomes could be combined with glycophorin A-coated vesicles, instead of lysosomes, and subsequently, be removed by exocytosis [48]. Our observations support this theory of organelle removal in EVs, as the RBC transcriptome was observed in EVs and especially enriched in mitochondrial transcripts and functions (oxidative phosphorylation, mitochondrial dysfunction), as well as transcripts for ribosomal proteins and translational machinery (eIF2 signaling). Whether RNA molecules are passively captured to EVs along with the removal of, e.g., unwanted proteins and organelles or if there might be some active selection, remains to be explored.

EV transcriptome contained a group of expressed genes not found in RBCs. They most likely originate from EVs of residual plasma in the RBC product resulting from the manufacturing process. Consequently, there is inherent biological variation between EV transcriptomes arising from small but variable amounts of plasma EVs. However, the overall EV transcriptome is similar to that of RBCs with RBC function-related transcripts expressed at a high level compared to contaminating ones indicating that vesiculation is truly a way to remove cellular components also from mature RBCs.

The most highly expressed transcript found in EVs but not in bulk RBC transcriptome, TMSB4X, was also found in some cells of the MALAT1 subpopulation and is previously associated with platelets [49]. However, it is unlikely that we have platelet contamination in the single-cell data since the same cells occasionally contained Hb and other RBC markers. In addition, cells were collected and purified in the same way for the bulk data, where platelet contamination was not detected. These transcripts could either have a very low expression in RBCs, hidden in the bulk data, and only become detectable in single-cell data in the cells that have lower Hb expression. If TMSB4X and TMSB10 transcripts are truly of platelet origin, one hypothesis is that platelet-EVs may have adhered to or even fused with [50] RBCs, thereby becoming a part of the RBC transcriptome. The transfer of transcripts via EVs is a well-known phenomenon [21].

## 4. Conclusions

Our study shows that RBCs still have a versatile transcriptome that can be utilized to produce not only Hb but also proteins needed in the final stages of reticulocyte maturation and for mature erythrocyte functions. The transcriptome of RBCs is transferred to EVs, which makes these transcripts available for intercellular communication in the blood. In addition, the RBC-EV transcriptome provides important background information for any plasma EV RNAseq studies and possible biomarker discovery.

Remarkably, we were able to determine the RBC transcriptome at the single-cell level, uncovering entirely new information on the final stages of RBC maturation. Single-cell sequencing of RBCs showed that there is heterogeneity among RBCs clustering in three different populations according to their transcriptomes. This data fits well with the accumulating understanding of the dynamic nature of reticulocyte maturation, making it unnecessary to strictly classify reticulocytes as one cell population. Recently, the feasibility of single-cell RNAseq of RBCs was confirmed by another study, where they showed the transcriptional heterogeneity of RBC subpopulations [51].

Our study revealed that the recently discovered lncRNA MALAT1 is the marker for one of the RBC populations co-expressing with a group of ribosomal protein transcripts. Our findings suggest a ribosomal location and a novel role for MALAT1 in translational control in reticulocytes. Moreover, we show that the relevant transcripts are not the full-length isoform MALAT1-202 but short, alternative transcripts currently only known as predictions. Functional studies are needed to verify the exact role of MALAT1 in maturing reticulocytes.

## 5. Materials and Methods

### 5.1. Isolation of RBCs, Extracellular Vesicles and RNA

Red blood cell (RBC) samples were collected on day 8 for bulk sequencing (n = 4) from carefully mixed leukoreduced RBC units (B+), stored in the saline-adenine-glucose-mannitol (SAGM) solution with citrate-phosphate-dextrose (CPD) anticoagulant at 2–6 °C under standard blood bank conditions (obtained from the Finnish Red Cross Blood Service, FRCBS, Helsinki, Finland) (experimental setup described in Figure 1). Two of the donors were female (aged 49 and 28) and two were male (aged 49 and 43). To remove any residual leukocytes, 10 mL of RBC unit was diluted with 10 mL of phosphate-buffered saline (PBS) and run through density gradient centrifugation (Ficoll-Paque plus, GE Healthcare, Chicago, IL, USA).

EVs were collected from the same RBC units. RBC concentrate sample (15 mL) was diluted with 15 mL of PBS and centrifuged first 20 min at 805× *g* and subsequently 20 min 3000× *g* without brake to remove the cells (Eppendorf centrifuge 5810R). The resulting supernatant was ultracentrifuged for 1 h at 100,000× *g* (+4 °C, fixed angle MLA-50 rotor, Beckman Coulter) to obtain EVs. Next, EVs were washed with PBS and ultracentrifuged for 1 h at 100,000× *g* (+4 °C, swing-out MLS-50 rotor, Beckman Coulter, Brea, CA, USA). Characterization of EVs obtained with this method has been described previously [52] and is shown in Appendix A.

For single-cell sequencing, one fresh blood sample (donor: female, 43 years) and one blood unit sample (donor male: 39 years), as comparable to fresh blood sample as possible (collected at day 2) were processed. RBCs from the fresh blood sample were collected similarly as from the blood unit sample and additionally purified with an Acrodisc WBC filter (Pall Life Sciences, Hoegaarden, Belgium).

Total RNA from RBCs (200 µL, ~1 × 10^9^ cells) and EVs was isolated by using the miRNeasy Mini Kit (Qiagen, Hilden, Saksa) according to the manufacturer’s protocol.

### 5.2. Preparation of RNAseq-Libraries and Sequencing

To prepare RNAseq libraries, 10 ng (RBCs) or 0.1–1 ng (EVs) of total input RNA was used with the SMART-seq v4 Ultra-low input RNA kit (Clontech, Takara Biotechnology Co., Ltd., Dalian, China). This method relies on poly-dT–based cDNA synthesis, which enabled us to avoid rRNA inserts in the libraries. RNAseq libraries were sequenced using the Illumina HiSeq 2000 system with v4 chemistry and 100 bp paired-end reads. Targeted sequencing depth was 20 × 10^6^ and 40 × 10^6^ paired-end reads for EV and RBC samples, respectively.

### 5.3. Analysis of RNAseq Data

R 3.4.0 was used to identify patterns in the RNAseq -data with Principal component analysis (PCA) and the Scatter plot function of Microsoft Excel was used to calculate correlation coefficients for data pairs originating from RBC and EV samples.

For data analysis, fastq data were trimmed with the Trimmomatic tool to remove Illumina platform-specific adapters. STAR –aligner was used to map the trimmed reads to the GRCh37r72 version of the human genome. Only reads uniquely mapped to the genome and not mapped to ribosomal RNA species were utilized for the HTseq tool to count reads mapping to genes. The RSeQC-tool package was used to determine the quality of RNAseq data.

Estimates for transcript abundances, reported as transcripts per million (TPM), were produced using the Kallisto algorithm [53] and further analyzed using the Sleuth software [54]. The final gene lists (minimum 5 counts/gene) from individual RBC and EV samples were compared using the Venny 2.1.0 tool [55] with Ensembl gene IDs.

### 5.4. Gene and Transcript Annotations, Pathway Analysis, and Functional Category Enrichment

Expression data were annotated, and the core expression analysis was performed using QIAGEN’s Ingenuity^®^ Pathway Analysis (IPA, QIAGEN, release Summer, 2018, Ensembl Human: GRCh38.86). Statistically significant enrichment of genes was examined on canonical pathways, molecular and cellular functions, and physiological system development and function -categories. Disease-related categories were omitted, except for the canonical pathway enrichment. On canonical pathways, all pathways having a *p*-value < 0.05 were listed. On functional categories, default settings were used, which resulted in the lists of categories with p values ≤ 0.00455 for EVs and ≤0.000168 for RBCs. The significance values were calculated by Fisher’s exact test righttailed. Ensembl IDs that could not be annotated with IPA were annotated using the Ensembl genome browser (release 88-Mar 2017).

### 5.5. Single RBC Analysis Using the 10X Genomics Platform

For single-cell analysis, RBCs were washed with 0.04% bovine serum albumin and 1 × PBS (Thermo Fisher Scientific, Waltham, MA, USA), filtered using a Flowmi Cell Strainer (40 µm, Bel-Art, Wayne, NJ, USA), counted with a LUNA Automated Cell Counter (Logos Biosystems, Gyeonggi-do, South Korea) and loaded into a Chromium Single-Cell Chip v2 (10× Genomics) for Chromium Controller gel beads in emulsion (GEMs) generation. The GEM generation, cDNA purification, and amplification were performed according to the manufacturer’s instructions using a Chromium Single-Cell 3′ v2 Reagent Kit with a 2000 target cell capture per sample. Library construction was performed according to the manufacturer’s instructions while scaling down the reaction. Sample libraries were sequenced in Illumina HiSeq 2500 using HiSeq Rapid SBS Kit v2 and HiSeq PE Rapid Cluster Kit v2 (Illumina, San Diego, CA, USA) according to the instructions by 10X Genomics.

The Cell Ranger v1.3 mkfastq and count analysis pipelines (10X Genomics, Pleasanton, CA, USA) were used to demultiplex and convert Chromium single-cell 3′ RNA-sequencing barcodes and read data to FASTQ files and generate aligned reads and gene-cell matrices. Blood unit and fresh RBC samples were analyzed as one sample. The Cell Ranger v1.3 aggr pipeline was used to produce a combined single gene-cell barcode matrix using normalization to the same sequencing depth. The gene-barcode matrices were analyzed using a Seurat 2.0.1 package [56,57] with R 3.4.0. First, the data were normalized to 10,000 molecules, and log-scaling was performed as described [56]. Genes detected in only one cell were excluded from the analysis. No further filtering was performed based on total unique molecular identifiers (UMIs) or the percentage of mitochondrial genes (Appendix A). For the remaining cells, principal component analysis (PCA) was performed using variable genes. First, ten principal components were used for clustering (resolution 0.3) and t-distributed stochastic neighbor embedding (t-SNE), allowing duplicates. Positive cluster marker genes were identified by comparing clusters using genes that were differentially expressed in at least 5% of the cluster’s cells and that showed at least a 0.10 log-scale fold difference on average between clusters. Genes with a *p*-value lower than 0.001 were considered significant.

The final gene list of the combined single gene-cell barcode matrix generated from fresh blood and blood unit was compared to bulk RNAseq data using Ensembl gene IDs and the Venny 2.1.0 tool. The comparison was restricted to genes with 5 or more total UMIs across the cells.

### 5.6. Visualization of Single-Cell RNAseq Data for MALAT1

An Integrative Genomics Viewer (IGV[58]) was used to visualize the aligned sequencing data (.bam files) with Ensembl Release 80 annotations, http://software.broadinstitute.org/software/igv/ (accessed on 22 October 2022), (Broad Institute, Cambridge, MA, USA).

### 5.7. Enrichment of Reticulocytes

In order to obtain enough RNA for qPCR, reticulocytes were enriched using CD71-conjugated microbeads (Miltenyi Biotec, Bergisch Gladbach, Germany) according to the manufacturer’s protocol. First, RBCs (n = 2 from blood unit, n = 3 from fresh blood) were purified as described above using density gradient centrifugation and an Acrodisc WBC filter (for fresh blood). Approximately 1 × 10^9^ RBCs were suspended in 800 µL of buffer (2 mM EDTA-PBS containing 0.5% BSA) and labeled with 150 µL CD71 microbeads. After 15 min incubation at room temperature, 10 mL buffer was added, and samples were centrifuged at 600× *g*. Samples were suspended in 800 µL of buffer and magnetic separation was done using MACS LS columns. After washing and elution, cells were counted using the Bürker chamber.

### 5.8. Analyses on the ImageStreamX MkII

To characterize the CD71-enriched samples (n = 5), the cells were labeled with CD71-PE and CD235a-Pacific blue antibodies (Biolegend, San Diego, CA, USA), as well as thiazole orange (TO) for RNA (Sigma, 0.01 ng/mL in PBS, stock diluted in DMSO) for 45 min at RT. Additionally, to check for platelet or leukocyte contamination, two samples were labeled with CD41a-FITC (BD Pharmingen, San Diego, CA, USA) and CD45-PerCp-Cyanine 5.5 (eBioscience, San Diego, CA, USA), respectively. At least 1000 events for each sample were acquired using a 12-channel Amnis ImageStream^X^ Mark II (EMD Millipore, Burlington, VT, USA) imaging flow cytometer. Samples were acquired at 60 × magnification with low flow rate/high sensitivity. The integrated software INSPIRE (EMD Millipore) was used for data collection. The instrument and INSPIRE software were set up as follows: Excitation lasers 405, 488, 642 and 785 and channels Ch1 and Ch9 (bright field, BF), Ch6 (scattering channel), plus fluorescence channels Ch3, Ch5, Ch7 and Ch11 were activated for signal detection. Single color controls were used for compensation. Compensated data files were analyzed using image-based algorithms available in the IDEAS statistical analysis software package. Single cells were separated from debris and doublets using a bivariate plot of aspect ratio vs. area of the BF image. Double positive events for CD71-PE and TO were gated as CD71^high^, CD71^medium^, and CD71^low-neg^ populations based on intensity values on each channel and image. In parallel, all samples were run with BD FACSAria IIU (BD Biosciences, Franklin Lakes, NJ, USA) flow cytometer using FACSDiva version 8.0.1 software (BD Biosciences) and analyzed with FlowJo version 10.0.7 software.

### 5.9. Quantitative Real-Time PCR

Quantitative real-time PCR (qPCR) was performed on CD71-enriched cells (n = 5). For cDNA synthesis, approximately 10^5^ cells were lysed and DNase I -treated using Taqman Cells-to-CT kit (Thermo Fisher Scientific, Waltham, MA, USA) according to the manufacturer’s instructions. The maximum amount of cell lysate (9 µL) per reaction was directly used for real-time qPCR with HBB (Hs00758889_s1) and MALAT1 Taqman assays. MALAT1 Taqman assay was custom-made against the sequence region found to be expressed in RNAseq analyses (Figure 6). Each reaction was performed in duplicate, and the signal was detected by ABI 7900HT real-time PCR machine. No template and –RT controls were included in the experiment. RNA could not be quantified and thus, MALAT1 and HBB expression levels are shown as quantitation cycle (Cq) values.

## Figures and Tables

**Figure 1 ijms-23-12897-f001:**
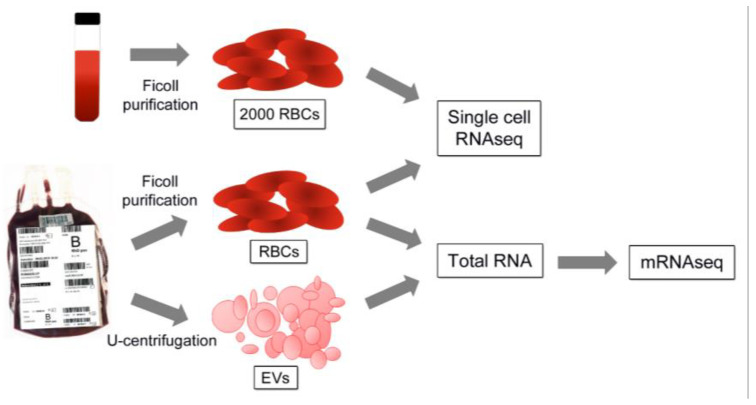
Experimental setup for bulk and single-cell sequencing. For bulk sequencing, RBCs from leukocyte-filtered blood units (n = 4) were purified with density gradient centrifugation and corresponding cell-free supernatant samples were ultracentrifuged to obtain EVs. For single-cell sequencing, a fresh blood sample (n = 1) and a sample from a blood unit (n = 1) were purified with density gradient centrifugation, as well as a fresh blood sample with white blood cell filtering. The total RNA was isolated and used for RNA sequencing.

**Figure 2 ijms-23-12897-f002:**
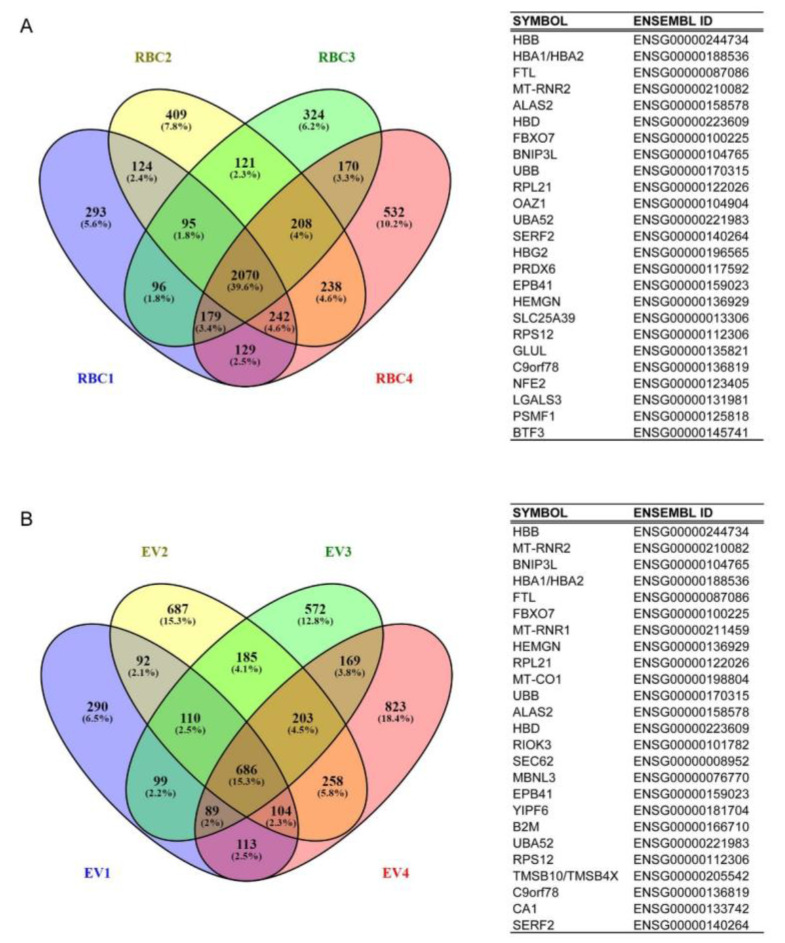
RBC (**A**) and EV (**B**) transcriptomes. Venny’s diagrams show the number of expressed genes for individual samples as well as those shared in all samples. The gene lists (minimum 5 counts/gene) from RBC and EV samples were compared with the Venny 2.1.0 tool using Ensembl gene IDs. On the right, the 25 most highly expressed genes are shown in the corresponding transcriptomes. The full lists with gene names and expression values are shown in Appendix A. Data was annotated using QIAGEN’s Ingenuity^®^ Pathway Analysis (IPA^®^, QIAGEN, 16 December 2016) or Ensembl genome browser (http://www.ensembl.org, release 88-Mar 2017, accessed on 3 April 2017).

**Figure 3 ijms-23-12897-f003:**
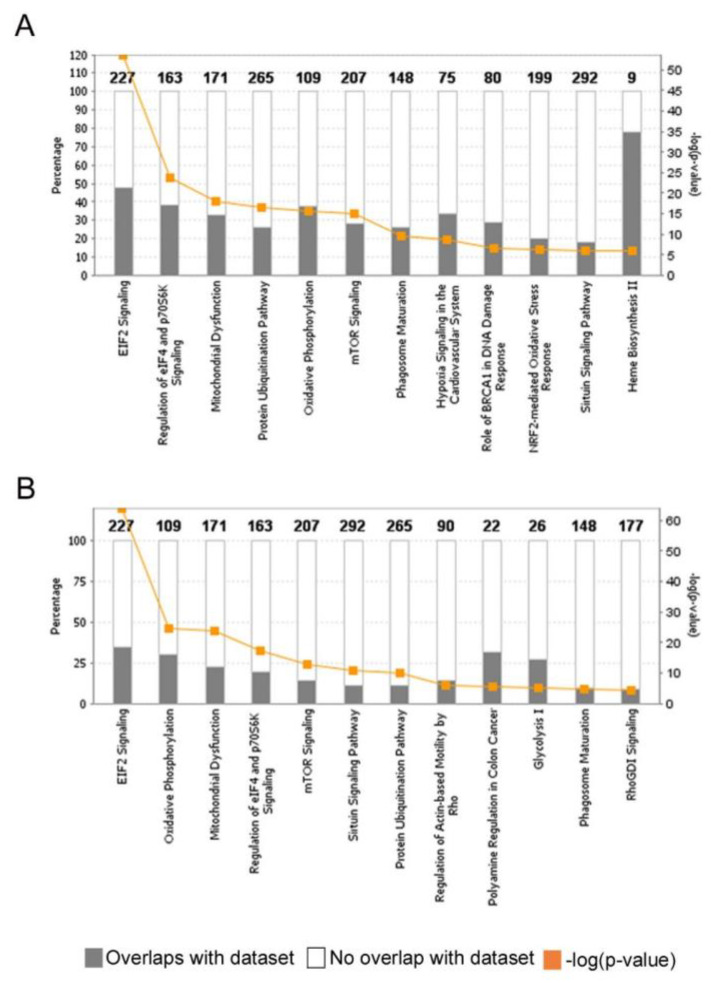
The 12 most significant canonical pathways in RBC (**A**) and EV (**B**) transcriptomes, as analyzed with QIAGEN’s Ingenuity Pathway Analysis (IPA). Overlaps and the -log(*p*-values, Fisher’s exact test right-tailed) are shown.

**Figure 4 ijms-23-12897-f004:**
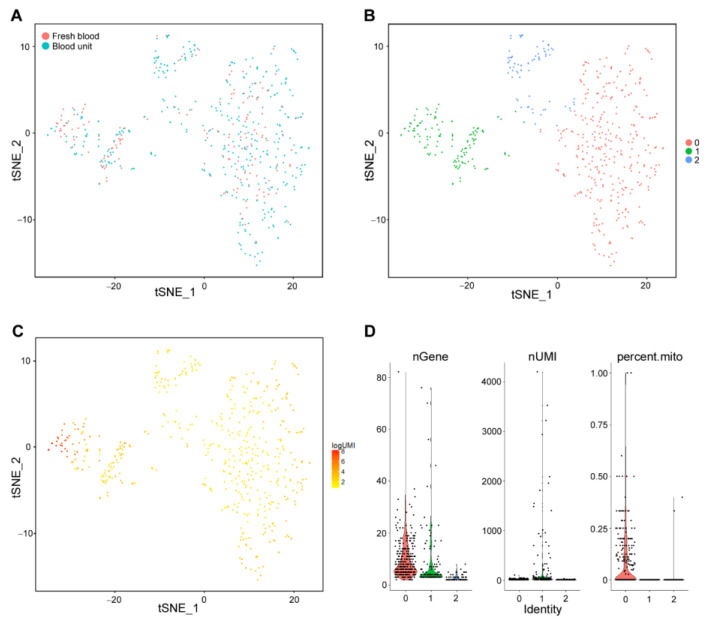
Single-cell analysis of RBCs reveals three distinct cell populations. (**A**) 2D t-distributed stochastic neighbor embedding (t-SNE) projection of 554 RBCs, with each cell colored based on their sample ID (blue; cells from blood unit, red; cells from the fresh sample. (**B**) The tSNE projection of 554 RBCs, where each cell is grouped into one of the identified 3 clusters (0, 1, 2). (**C**) tSNE projection of 554 RBCs, with each cell colored based on their total unique molecular identifier (UMI), counts on a logarithmic scale. (**D**) Violin plots showing the distribution of the total number of genes (nGene), total UMIs (nUMI), and the percentage of mitochondrial gene expression (percent.mito) per cell for the identified three RBC clusters.

**Figure 5 ijms-23-12897-f005:**
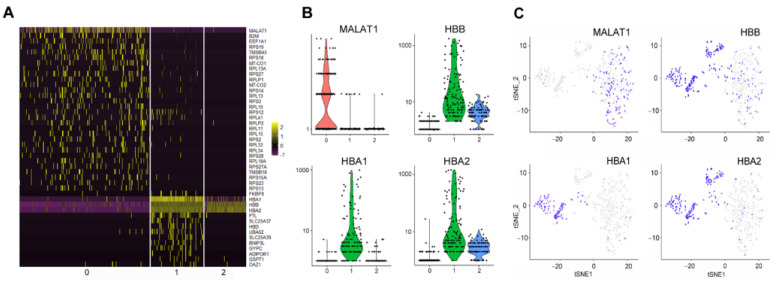
Three RBC populations are characterized by specific marker gene expression profiles. (**A**) A heatmap showing the top 30 marker genes for each RBC cluster. The expression of each gene is normalized against the total expression within the cell. (**B**) Violin plots showing the expression of MALAT1, HBB, HBA1 and HBA2 genes as the unique molecular identifier (UMI) counts per cell. (**C**) T-distributed stochastic neighbor embedding (tSNE) projection of 554 RBCs, with each cell colored based on its expression of MALAT1, HBB, HBA1, and HBA2 genes. Blue color indicates a positive expression, and gray indicates a negative expression.

**Figure 6 ijms-23-12897-f006:**
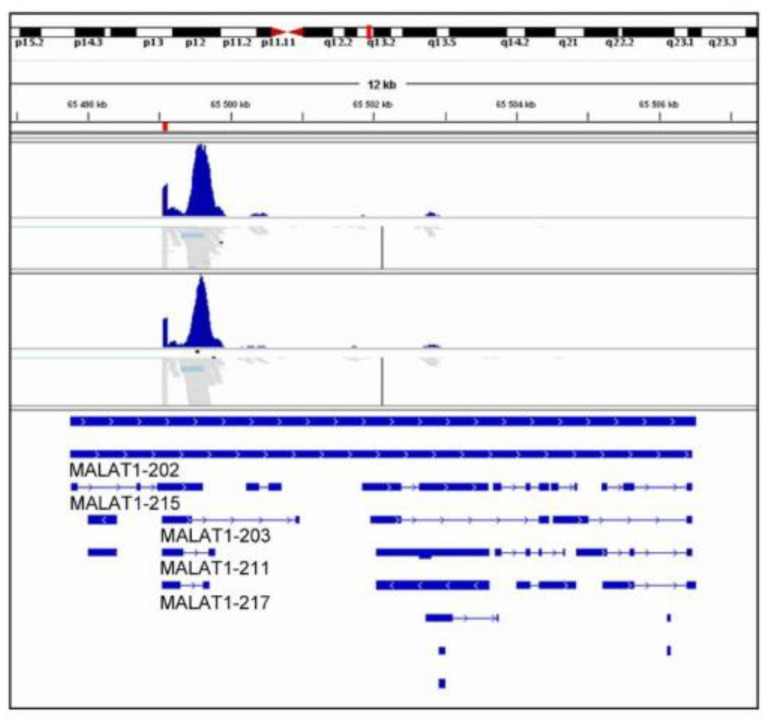
Alignment of sequencing reads mapping to MALAT1 gene from single-cell RNA sequencing analysis. The alignment was performed with an integrative genomic viewer (IGV) using Ensembl gene annotations to visualize short reads matching the MALAT1 gene.

**Figure 7 ijms-23-12897-f007:**
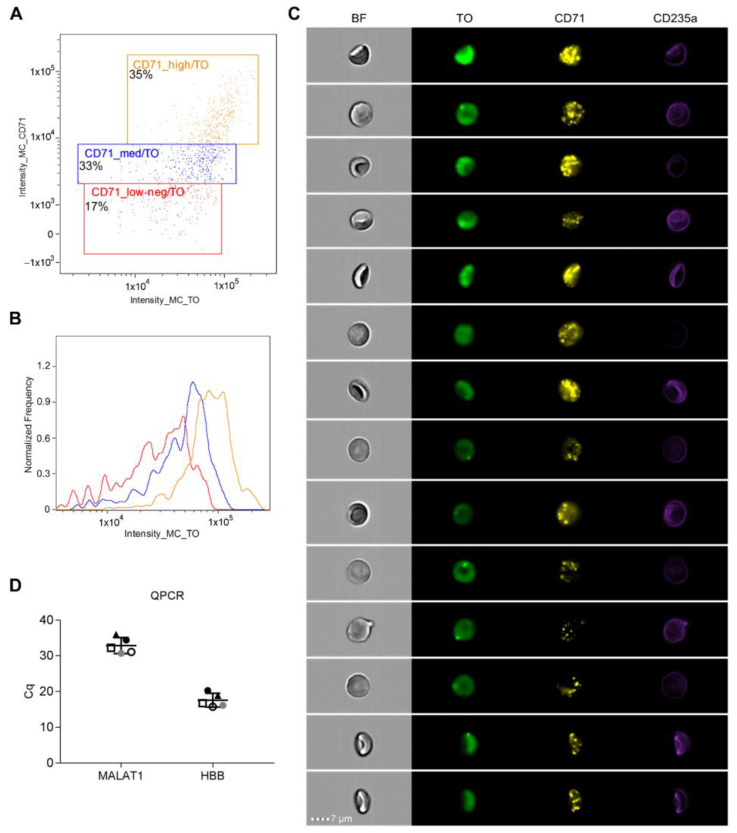
Characterization of enriched reticulocytes using imaging flow cytometry and Q-PCR. Imaging flow cytometry was performed for CD71-enriched cells, stained with CD71-PE and CD235a-PB antibodies and thiazole orange (TO) for RNA content. (**A**) A dot plot image showing double positive cells for CD71 and TO, as well as gates for CD71_high, CD71_medium, and CD71_low-neg populations from a representative sample. (**B**) An overlay of flow cytometry histograms demonstrates the intensity range of TO RNA stain in different CD71 populations. (**C**) Representative image gallery showing different morphologies and staining patterns for CD71-enriched cells. Each cell is represented by a row of four images acquired simultaneously in flow, from left to right: brightfield (BF), the RNA binding dye TO (green), CD71-PE (yellow), CD235a-Pacific Blue (violet). Scale bar 7 µm. (**D**) The expression of the MALAT1 and HBB genes in the CD71-enriched samples (n = 5) is shown as quantification cycle (Cq) values. Lines depict the mean ± SD of all replicates. Corresponding samples are illustrated with matching symbols.

**Table 1 ijms-23-12897-t001:** The most enriched pathways and functions in red blood cell (RBC) and RBC-derived extracellular vesicle (EV) transcriptomes. The top five categories are shown in each group, as well as *p*-values or *p*-value ranges.

RBC Transcriptome	
Canonical Pathways	*p*-value
EIF2 Signaling	2.53 × 10^−54^
Regulation of eIF4 and p70S6K Signaling	1.65 × 10^−24^
Mitochondrial Dysfunction	1.05 × 10^−18^
Protein Ubiquitination pathway	2.77 × 10^−17^
Oxidative Phosphorylation	1.78 × 10^−16^
**Molecular and Cellular Functions**	***p*-value range**
Gene expression	5.17 × 10^−31^–2.57 × 10^−7^
Protein synthesis	4.81 × 10^−30^–7.75 × 10^−5^
Cell Death and Survival	1.47 × 10^−25^–1.68 × 10^−4^
RNA Post-Transcriptional Modification	7.82 × 10^−25^–2.78 × 10^−5^
Cell Cycle	1.86 × 10^−16^–1.43 × 10^−4^
**Physiological System Development and Function**	***p*-value range**
Organismal Survival	4.93 × 10^−14^–4.39 × 10^−6^
Connective Tissue Development and Function	4.00 × 10^−9^–5.15 × 10^−5^
Hematological System Development and Function	4.25 × 10^−9^–4.53 × 10^−5^
Hematopoiesis	4.25 × 10^−9^–4.53 × 10^−5^
Tissue Morphology	4.25 × 10^−9^–4.53 × 10^−5^
**EV transcriptome**	
**Canonical Pathways**	***p*-value**
EIF2 Signaling	1.14 × 10^−64^
Oxidative Phosphorylation	3.21 × 10^−25^
Mitochondrial Dysfunction	1.60 × 10^−24^
Regulation of eIF4 and p70S6K Signaling	2.90 × 10^−18^
mTOR Signaling	1.68 × 10^−13^
**Molecular and Cellular Functions**	***p*-value range**
Cell Death and Survival	2.92 × 10^−31^–4.49 × 10^−3^
Protein synthesis	8.21 × 10^−24^–4.27 × 10^−3^
Gene expression	3.53 × 10^−19^–2.95 × 10^−3^
RNA Post-Transcriptional Modification	4.11 × 10^−13^–7.87 × 10^−4^
Free Radical Scavenging	3.24 × 10^−9^–2.28 × 10^−3^
**Physiological System Development and Function**	***p*-value range**
Hematological System Development and Function	6.36 × 10^−7^–3.56 × 10^−3^
Connective Tissue Development and Function	1.10 × 10^−6^–4.55 × 10^−3^
Hematopoiesis	4.02 × 10^−6^–4.07 × 10^−3^
Tissue Morphology	4.02 × 10^−6^–1.47 × 10^−3^
Organismal Development	7.03 × 10^−6^–2.28 × 10^−3^

Statistically significant overrepresentation of genes was examined on canonical pathways, molecular and cellular functions and physiological system development and function categories with QIAGEN’s Ingenuity Pathway Analysis (Summer, 2018). The significance values are calculated by Fisher’s exact test right tailed.

## Data Availability

All sequencing data were deposited in NCBI’s Gene Expression Omnibus [59], and they are accessible through the following GEO Series accession numbers: bulk RNAseq -data GSE108378 (https://www.ncbi.nlm.nih.gov/geo/query/acc.cgi?acc=GSE108378), and single-cell RNAseq data GSE108241 (ongoing process, this link will be delivered later).

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
