# Peer review of "Exploring Transcriptomic Landscapes in Red Blood Cells, in Their Extracellular Vesicles and on a Single-Cell Level"

_ijms, 2022, doi:10.3390/ijms232112897_

Round 1

Reviewer 1 Report

The manuscript from Kerkela et al. reports on the analysis of RBCs and RBC-derived EVs by next-generation sequencing. The paper is technically sound and well written. Only some minor remarks:

- The Materials and Methods section should be placed either after the introduction or at the end of the manuscript (i.e., after the conclusions section).

- On page 7, the authors mention 3 clusters, but in the Discussion (page 12.) they mention "Two distinct subpopulations", which is a bit confusing. Furthermore, can the clustering of the RBC based on the single-cell RNAseq and based on the flow cytometry experiments be correlated? 

- In the end, how does the transcriptome of RBCs distribute among young and mature reticulocytes and erythrocytes? And how does the transcriptome of RBCs compares to other blood cells? An extended discussion would help to non-expert readers (such as the reviewer).

Author Response

The authors sincerely thank the editor for the opportunity to re-submit the manuscript after revisions, and the reviewers for the valuable comments on the manuscript. We have reviewed the comments and suggestions in detail and have now addressed all comments and revised the manuscript according to reviewers’ suggestions. Changes in the text are highlighted with ‘track and change’. 

General comments:

We added one reference to the ‘Conclusion’ section, page 13, row 388. The paper was recently published and confirms the feasibility of single cell RNAseq from RBCs and thus supports our work. This is an important paper to be added to the manuscript: Jain V, Yang WH, Wu J, Roback JD, Gregory SG, Chi JT. Single Cell RNA-Seq Analysis of Human Red Cells. Front Physiol. 2022 Apr 20;13:828700.

We have carefully checked the spelling and technical errors in the text. I hope it is now more readable. English editing was not done due to short time to revise and respond.

Reviewer1

The manuscript from Kerkela et al. reports on the analysis of RBCs and RBC-derived EVs by next-generation sequencing. The paper is technically sound and well written. Only some minor remarks:

- The Materials and Methods section should be placed either after the introduction or at the end of the manuscript (i.e., after the conclusions section).

Response: We sincerely than the reviewer of the kind comments and this suggestion has now been done. ‘The Materials and Methods’ section has now been placed to the end and ‘Conclusions’ after ‘Discussion’.

- On page 7, the authors mention 3 clusters, but in the Discussion (page 12.) they mention "Two distinct subpopulations", which is a bit confusing. Furthermore, can the clustering of the RBC based on the single-cell RNAseq and based on the flow cytometry experiments be correlated? 

Response: Yes, three distinct subpopulations were identified. This is also stated at the discussion, first sentence of page 12: “Based on our single-cell analysis, there are significant differences in the transcriptomes of three RBC subpopulations.” It is just further defined that two of them are more alike so two clusters contain globin genes while is defined by MALAT1 expression: “Two distinct subpopulations are defined by the expression of globin genes, and one is defined by the expression of the lncRNA, MALAT1.”. I now rewrote the sentence so that it is more clear to the reader.

Unfortunately, although the correlation of single-cell RNAseq clustering vs. the flow cytometry experiments/data would be very interesting, it is impossible as the data is, first of all, so different from each other and not generated from the same cells. Division of flow cytometry data (double positive events for CD71 and RNA dye thiazole orange) gated as CD71high, CD71medium and CD71low-neg populations was based on expression intensity values on each channel and images. Division is still arbitrary made by a researcher and not similarly done as single cell RNAseq data.

- In the end, how does the transcriptome of RBCs distribute among young and mature reticulocytes and erythrocytes? And how does the transcriptome of RBCs compares to other blood cells? An extended discussion would help to non-expert readers (such as the reviewer).

Response: This is indeed a relevant question. However, we cannot say for sure that how the transcriptome is divided between different RBCs or reticulocytes, since we have no way to identify the subpopulations other than the transcriptome itself. It is just speculation that as reticulocytes still have some active translation from mRNA, it would be logical that the younger the reticulocytes the more mRNA they would contain. This seem to be the case based on the overall correlation of CD71 marker and RNA content (stained with TO) in flow cytometry. Cells would then gradually lose the RNA content as they stop translation and become mature erythrocytes. So it could be both the amount and diversity of mRNA that would define the age of reticulocytes, but we cannot, unfortunately, say for sure.

When we compare RBCs to other blood cells, RBCs are enucleated so their transcriptome differs a lot from other blood cells. They cannot synthetize new mRNAs after enucleation, i.e., in blood circulation. So, it is both the diversity and the amount of mRNA that is different from white blood cells. Transcriptome reflects the functionalities of RBCs, as described in ‘Results’ section quite extensively. Moreover, the uniqueness of RBC transcriptome is described in the ‘Introduction’. We hope that it is enough for the general differences between RBCs and other blood cells. Even though the discussion of the differences would be relevant, it is difficult to fit in the ‘Discussion’ section which is now from different viewpoint. Differences hopefully come out generally from the whole text and from the fact that RBCs are enucleated cells.

Reviewer 2 Report

In this manuscript, the authors explored the transcriptomes of RBCs and extracellular vesicles (EVs) of RBCs using next-generation sequencing. To better clarify these aspects, the authors studied the transcriptomes of single-cell RBCs from both fresh blood and RBC unit. The scientific work is very interesting, minor problems, as indicated below, should be addressed before the document can be considered for the publication. 

Minor revision:

-The paper is clearly written and complete in most respects. English language and style are fine, minor spell check required. Moreover, some typing errors are presented in the full text.

-Line 67. Similar vesiculation occurs during storage-related aging. To improve this evidence, I suggest to add a new recent manuscript (doi: 10.3390/cells11152391).

-The resolution of all Figures should be improve. 

Author Response

The authors sincerely thank the editor for the opportunity to re-submit the manuscript after revisions, and the reviewers for the valuable comments on the manuscript. We have reviewed the comments and suggestions in detail and have now addressed all comments and revised the manuscript according to reviewers’ suggestions. Changes in the text are highlighted with ‘track and change’. 

General comments:

We added one reference to the ‘Conclusion’ section, page 13, row 388. The paper was recently published and confirms the feasibility of single cell RNAseq from RBCs and thus supports our work. This is an important paper to be added to the manuscript: Jain V, Yang WH, Wu J, Roback JD, Gregory SG, Chi JT. Single Cell RNA-Seq Analysis of Human Red Cells. Front Physiol. 2022 Apr 20;13:828700.

We have carefully checked the spelling and technical errors in the text. I hope it is now more readable. English editing was not done due to short time to revise and respond.

Reviewer2

In this manuscript, the authors explored the transcriptomes of RBCs and extracellular vesicles (EVs) of RBCs using next-generation sequencing. To better clarify these aspects, the authors studied the transcriptomes of single-cell RBCs from both fresh blood and RBC unit. The scientific work is very interesting, minor problems, as indicated below, should be addressed before the document can be considered for the publication. 

Minor revision:

-The paper is clearly written and complete in most respects. English language and style are fine, minor spell check required. Moreover, some typing errors are presented in the full text.

-Line 67. Similar vesiculation occurs during storage-related aging. To improve this evidence, I suggest to add a new recent manuscript (doi: 10.3390/cells11152391).

-The resolution of all Figures should be improve. 

Response: We thank the reviewer for the kind comments on our manuscript. We have carefully checked the spelling and technical errors in the text. I hope it is now more readable.

We have also added the fresh reference about the vesiculation. The ref. numbering has been changed accordingly.

Figure 1 was done again to improve the resolution and added to the manuscript. Figure 3 we could not unfortunately redo, as we do not have access to the software anymore. Otherwise we have tried to improve the resolution of images, although somehow the word format is not the best in maintaining the good resolution of images.